# From self-interest to collective action: The role of defaults in governing common resources

Eladio Montero-Porras[1,2,3*], Rémi Suchon[4],
Tom Lenaerts[1,2,3,5], Elias Fernández Domingos[1,2,3]

**1** Artificial Intelligence lab, Vrije Universitet Brussel, Brussels, Belgium, **2** Machine Learning Group, Université Libre de Bruxelles, Brussels, Belgium, **3** FARI Institute, Universite Libre de Bruxelles-Vrije Universiteit Brussel, Brussels, Belgium, **4** Anthropo-Lab-ETHICS, Université Catholique de Lille, Lille, France, **5** Center for Human-Compatible AI, University of California, Berkeley, California, United States of America

* eladio.montero.porras@ulb.be

**Data availability statement:** The source dataset, the code used for the analysis and to

## Abstract

Managing shared resources requires balancing personal profit and sustainability. This paper reports on a behavioural experiment testing how extraction defaults—either pro-social or exploitative—impact resource extraction in a common pool resource dilemma (CPRD). We find that an exploitative default increases average extraction compared to a control without a default, while a pro-social default temporarily reduces extraction. The effects of both defaults are temporary, and extraction levels converge to those in the control group, with the pro-social default fading faster. Notably, the influence of defaults depended on individual inclinations, with cooperative individuals extracting more under an exploitative default, and selfish individuals less under a pro-social default. Our findings suggest that while defaults can promote short-term sustainability, their long-term effects are limited, and their effectiveness depends on individual traits.

## 1 Introduction

Modern life relies heavily on finite resources like water, electricity, internet bandwidth or public roads. The over-consumption of these resources may produce negative externalities on society and the environment. These pose significant challenges and are difficult to overcome, as individual rationality may easily lead to over-consumption, resulting thus in a "tragedy of the commons" [1]. In consuming the commons, people might cooperate by limiting their water usage during droughts or by sticking to agreed limits in shared grazing areas, thereby helping preserve the resource for all. Conversely, others may free-ride by consuming as much as they wish or by overexploiting shared fisheries, assuming that others' restraint will be sufficient to prevent depletion [2]. Resolving such social dilemmas require the design of institutions and mechanisms to enforce rules. A central authority, e.g. the state, can impose sanctions and rewards (such as taxes, or extraction rights) to tame over-consumption. In opposition to such a top-down approach, Elinor Ostrom's seminal work [3] demonstrates the

reproduce the figures can be found in Zenodo https://doi.org/10.5281/zenodo.10818839.

**Funding:** E.M.P and T.L. benefit from the support by the Flemish Government through the AI Research Program and by TAILOR (https://tailor-network.eu/), a project funded by the EU Horizon 2020 research and innovation program under GA No 952215. T.L. is furthermore supported by the F.N.R.S. (https://www.frs-fnrs.be/fr/) projects with grant number 31257234 and 40007793, the F.W.O. (https://www.fwo.be/nl/) project with grant no. G.0391.13N, and the Service Public de Wallonie Recherche (https://recherche.wallonie.be/home. html)https://recherche.wallonie.be/home.html) under grant no. 2010235–ARIAC by DigitalWallonia4.ai. E.F.D is supported by an F.N.R.S (https://www.frs-fnrs.be/fr/) Chargé de Recherche position, grant number 40005955. The sponsors or funders did not play any role in the study design, data collection and analysis, decision to publish, or preparation of the manuscript.

**Competing interests:** The authors have declared that no competing interests exist.

effectiveness of decentralised, bottom-up arrangements in preserving exhaustible resources through self-governance.

The consumption and regulation of many essential, finite resources occurs through either service providers (e.g., water and energy suppliers) or governmental institutions (e.g., natural resource management agencies, environmental protection departments). These mediating entities can play a decisive role in curbing over-consumption, by encouraging clients/consumers towards more desirable consumption behaviours. For example, consumers can be incentivised to adopt more sustainable food choices [4–6], put more money aside for savings [7,8], consume from more eco-friendly sources [9,10], and to reduce their overall resource consumption [11,12].

Consumption choices are also affected by an increasingly complex choice landscape, with many sources of information and numerous options. Intelligent automated systems offer the promise and opportunity to ease and optimise these choices [13]. An important factor among these decision-support systems is that their design, and consequently the choice architecture, is determined by the mediating entities mentioned earlier. For example, it is common for energy providers to offer their clients default consuming plans. These default options, not only make it easier for consumers to make their desired choices, but can potentially play a role in promoting the use of 'green' energy sources [14]. Similarly, Berger et al. (2022) find that setting carbon offsets as the default in flight bookings significantly increases voluntary climate action, even when the cost amounts to several hundred euros [15].

Users may respond differently to the available default setting, influenced by their own inclinations and priorities [16]. In this line, Behlen et al. (2023) found that defaults, while being effective, require a targeted approach to reach individuals whose interests align with the policy-maker [17]. For instance, an electricity provider can offer renewable energy for consumers by default, promoting a course of action supported by policy or a prevailing social norm. Yet, some consumers may prefer other sources of energy and adapt their consumption habits to accommodate their individual preferences [18,19]. From the perspective of libertarian paternalism [20], one strength of these mechanism is the ability to override such defaults, as it allows individuals to act on their own preferences. People who are particularly insensitive to the collective issue of energy over-consumption may be more likely to override a default directed towards renewable energy sources [21]. However, one concern is that not enough people opt out even when the default does not match their preferences [22]. So far, no clear answers have been provided experimentally on this association between personal preferences, default settings and consumption of a finite resource. In this work, we aim to address this gap in the literature by examining this question.

First, we want to assess whether a simple manipulation, such as setting a default extraction value, can curb collective (over-)consumption in a Common Pool Resource dilemma (CPRD) [3,23]. The CPRD is particularly well-suited to study consumption of a finite resource: CPRD experiments have been used to study the exploitation of water basins, lakes, irrigation of a community, fisheries or timber, to name a few [24–27]. The problem lies in the aggregate behaviour of participants: without a system of governance, participants excessively appropriate from the common resource, which may give rise to a "tragedy of the commons" [28]. See Fig 1A for a visual representation of the CPRD.

A meta-analysis by Mertens et al. (2022) demonstrated that choice architecture strategies, particularly those on decision structure like default settings, often surpass others focusing on decision information or assistance [29], revealing the potential efficacy of such interventions. Although no evidence of nudging was found when this meta-analysis is adjusted for publication bias [30]. Fosgaard et al. (2015) observed that in public goods games, presenting a conditional cooperation strategy as the default option effectively nudged participants

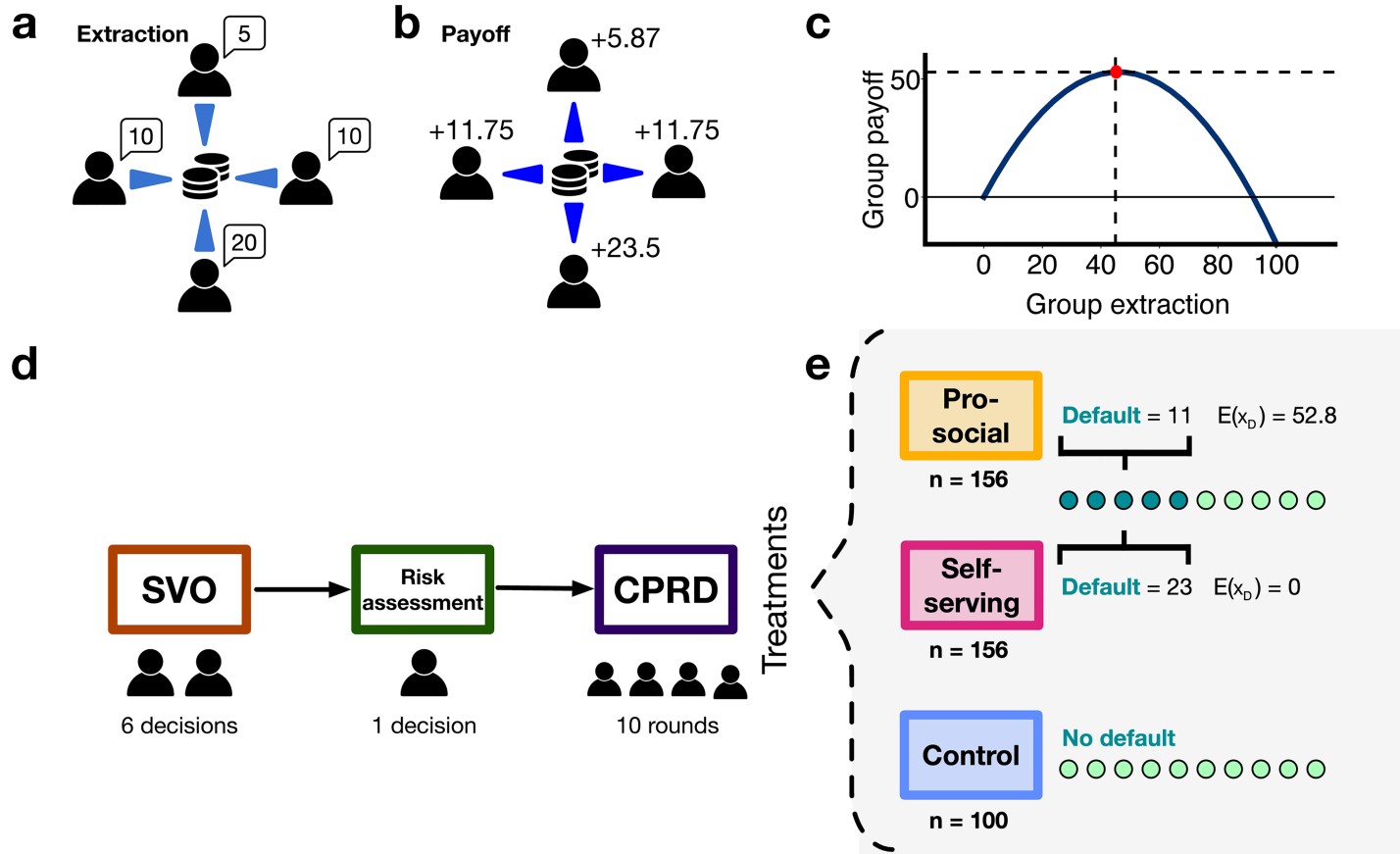

**Fig 1. Visual representation of the experiment designed for this work.** We use the Common Pool Resource dilemma (CPRD) [23] to understand collective resource management. In this game, individuals extract resources (as tokens) and receive payoffs (in experimental units, or ECUs) proportionate to their extraction levels. Panel **A** shows four individuals who request 45 tokens in total. Panel **B** transforms these requests into a collective payoff of 52.87 ECUs, determined by a payoff function curve shown in panel **C**. (more details in the Methods section). **D:** Experiment flow. the experiment consists of three tasks: In task 1 (brown box), participants complete an incentivised Social Value Orientation task (SVO) [50], allowing us to identify participants' resource allocation preferences. Task 2 (green box), assessed participants' risk aversion (also incentivised) [51,52]. Finally, in task 3, participants engage in the CPRD as part of one of three treatments, which cover 2 different types of defaults and one treatment without a default for comparison (panel **E**): Pro-social (default = 11 tokens, optimal group extraction yielding 52.8 ECUs), Self-serving (default = 23, excessive extraction yielding no returns), and a Control with no preset extraction. The green dots represent the rounds, where the darker green represent the first five rounds with default values, and the lighter green the rounds with no default. More details on these treatments are provided in the Methods section.

towards more cooperative behaviour [31]. Similarly, Bynum et al. (2016) studied how default non-participation in collective risk dilemmas (contributing zero by default) led to participants more frequently failing to meet the required threshold in this game [32]. Moreover, Ferguson et al. (2020) found in their dynamic organ donor game that default opt-out decisions for non-cooperators significantly impact group cooperation levels, more so than opt-in decisions for cooperators [33]. Furthermore, default settings significantly impact environmental choices. Liebe et al. (2021) showed that when sustainable, or 'green', energy options are set as the default, their adoption rates increase, resulting in reduced energy consumption [19]. Conversely, when 'grey' energy options, which are less eco-friendly, are defaulted, participants adopted these less sustainable sources more often [34]. This study builds on prior research by evaluating defaults in the CPRD management, contrasting a socially beneficial default with a

socially detrimental one in terms of their influence on both increasing and reducing resource consumption, particularly in relation to overextraction.

Second, we aim to understand how default settings impact long-term consumption habits, and what happens when the default is lifted. In our experiment, we presented the default to participants in the first five rounds and this pre-selected value disapears from rounds six to ten. We do this to assess the potential for both positive and negative spillovers. Positive spillover refers to scenarios where a pro-social behaviour leads to more of such behaviour, enhancing thus sustainability efforts. Conversely, negative spillover, or backfire effects, occur when an initial pro-social action is followed by opposing behaviours (see the work of Truelove et al. (2014) discussing these spillover effects in detail [35]).

In this regard, the experimental findings are mixed. Cappelletti et al. (2014) found a decline in choice of the default contribution following the removal of defaults in a public goods game, highlighting the challenges in sustaining behaviours when the default is removed [36]. Fosgaard et al. (2015) found that after participants saw a free-rider strategy by default, they were significantly more defective in subsequent public goods game without defaults [31]. Manganari et al. (2022) found that opt-out defaults (such as those used in the present study) are more effective at changing consumer behaviour, whereas opt-in defaults (which require users actively to select the option) lead to more enduring adherence [37]. Chaudhuri and Paichayontvijit (2017) found cooperation decayed after discontinuing exhortative messages in the Public Goods Game, though remaining above baseline [38], while Fehr and Gächter (2000) observed contributions immediately collapsed when punishments were eliminated [39]. However, contrasting these findings, Ghesla et al. (2019) found that encouraging pro-social decisions through choice defaults, with or without significant opt-out costs, does not affect unrelated subsequent pro-social behaviour [40] and they might be effective for subsequent tasks [41]. Albeit these soft interventions are very popular for policy-making, their long-term efficacy has been questioned [42]. Our work expands on this current understanding by testing the presence and nature of spillover effects when default values are lifted.

Lastly, our study examines the dual nature of default effects by introducing both pro-social and self-serving defaults, to assess their influence on group decisions towards either beneficial or harmful outcomes. Therefore, the third and final goal is to explore how the impact of defaults on (over-) consumption is mediated by (the heterogeneity of) individual preferences. We introduce two different default values, which might encourage pro-social individuals towards increased extraction and individualistic ones towards reduction. Understanding the heterogeneous effect of nudges could help tailor nudge interventions to individual characteristics and make them (more) effective [17,43]. The effectiveness of defaults is moderated by personality traits such as neuroticism and extraversion [37], as well as by individual differences in anxiety and avoidance tendencies [44]. However, mismatches between these two can lead to backfiring, surpassing the impact of transparent information or modes of thinking [45]. Guido et al.'s (2023) experiment showed that rule-followers responded more to nudges with persuasive, socially conscious messages than rule-breakers, but no average effect was noted when social preferences were ignored [46]. Additionally, Ghesla et al. (2017) found in the context of energy consumption that politically left-leaning individuals or those prioritising environmental concerns are likelier to opt out of default gray energy contracts [47]. We build on these findings by examining participants' social preferences and their responses to the default, as social preferences are key to understanding the effectiveness of cooperation interventions [48].

To achieve our three goals, we introduce a behavioural experiment using as the main task the Common Pool Resource Dilemma (CPRD) [23]. This game is widely used to model the dynamics of finite resource consumption [3], both in and outside the laboratory [49].

In the CPRD experiment, as represented in Fig 1, participants in groups of four have to coordinate their consumption of a finite resource, which offers benefits to consumers based on their proportional usage. They repeat this task for 10 rounds. Two default scenarios are presented: i) the *Self-serving* default, while enticing since it leads to higher payoff if the rest extract reasonably, selecting this value results in zero benefits if it is chosen by everyone in the group; and ii) the *Pro-social* default, which, if collectively adopted, yields the social optimum, maximizing the resource's potential benefits (see Methods for details). Although the default value in the CPRD suggest a course of action, participants can override it and choose another consumption level (from one to a maximum of 30 tokens). This preserves the participants' autonomy over the shared resource. To assess the ability of a default to build lasting consumption habits, our second goal, we provide the default only during the first five rounds of the experiment, and remove it in the last five.

To map social preferences and risk aversion to the effect of defaults in the CPRD, participants are asked to complete two behavioural tests before the main experiment starts: a Social Value Orientation (SVO) test [53], and a risk-aversion test [54] (see Fig 1). We use a SVO test to classify participants into four social-preference types: altruistic, pro-social, individualistic and competitive individuals. Pro-social individuals, identified by their SVO, have been shown to exhibit heightened concern for environmental causes and collective welfare [55–57]. Research has demonstrated that individuals classified as individualistic and competitive tend to extract significantly more than those classified as altruistic or prosocial in resource dilemmas [58]. Moreover, certain decisions can pose greater challenges for specific individuals, such as the relatively slower evidence collection among equality seekers in the Prisoner's Dilemma (PD) (players using reciprocal strategies such as Tit-for-Tat) compared to categorical decision-makers (player unconditionally cooperating or defecting in the PD) [59]. Lastly, we also measure the level of risk-aversion of participants and correlate it with default adoption, as the default option is often perceived as the safe and risk-free option [54]. Moreover, we want to test the relationship between risk perception and consumption, since risk-averse individuals have shown to consume less in CPRD experiments [60].

In summary, in this work we aim to answer three main research questions: 1) Is the presence of a default extraction influencing participants extractions in the CPRD to be higher or lower compared to having no default? 2) Does the default effect persist if the default value is no longer presented to participants? and 3) What is the effect of setting a default to shift from self-serving social preferences towards cooperation? Conversely, what is the effect of setting a default to shift from pro-social social preferences towards over-consumption?

## 2 Methods

### 2.1 Study design

The data used in this paper was collected through a behavioural experiment, and the participants were collected through the online participant pool Prolific (www.prolific.com) from May 8th, 2023 until August 28th, 2023. In Prolific, we required participants who indicated they were fluent in English, had a high approval rate (+99%) and had more than 20 submissions on the site. This was done to avoid dropouts. (see Supporting Information Sect S0.2 for more details about the participants' metadata). All participants were required to provide written informed consent before starting the experiment, and all participants were aged 18 or older. Participants had to complete three tasks (see Fig 1). Task 1 is the Social Value Orientation test (SVO) from Murphy et al. (2011) [61], and task 2 is the Risk-attitude elicitation task developed by Eckel and Grossman [52] with the values used by Dave and Eckel [51] see Sect 2.2 for more details on what we ask participants on each of these tasks. In task 3, participants

play a CPRD in a group of 4. We designed the experiment so we can link extraction behaviour in the CPRD in task 3 with participants' social preferences and risk attitudes elicited in tasks 1 and 2. Our experimental design, hypotheses, analyses sample size were pre-registered in OSF: https://osf.io/jg2sa. Additionally, we included the list of deviations from this pre-registration in the analysis perfomed in this work. Find the list of changes in Sect S0.1.1.

**2.1.1 Common pool resource dilemma.** In the main task, task 3, participants played a version of the Common Pool Resource Dilemma (CPRD). This dilemma captures the tension that emerges in a group exploiting a common finite resource. The group payoff is determined by the sum of extractions of the group members: over-extraction depletes the resource, leading to a payoff of zero to everyone, while a sustainable management of the resource yields the highest collective payoff. Individual payoff is proportional to individual extraction: one gets a higher share of the group payoff if one extracts a bigger share of the group extraction. Hence, narrow self-interest dictates to appropriate the resource as much as possible, however, if everyone does so, depletion ensues (tragedy of the commons), which is the worst collective outcome.

We propose a variant of the game where participants have to extract a minimum of 1 and maximum 30 units (or tokens, as they are called in the experiment) from the resource, which participants can select from a drop-down menu in each round. The extraction variable is discrete, that is, participants were able to extract any natural number in the interval $[1..30]$. The CPR replenishes at every round meaning that their past actions did not affect how much participants can extract at a given round.

The amount extracted by each player $i$ from the resource is $x_i$, and the amount extracted by the group is $X = \sum_{i=1}^{N} x_i$. Extraction of the resource earns each player $a$ times every unit extracted personally, minus $b$ for every unit extracted by the group regardless of who extracts it. We adjusted the parameters used by Walker et al. (2019) [23] to obtain similar payoffs among the different tasks ($a = 2.3$ and $b = 0.025$). The payoff of extracting $x_i$ and the group extraction $X$ from the CPRD is then:

$$\pi_i = (x_i)(a - bX) \tag{1}$$

Participants have the following information at all times (see Fig 2):

1. The round number.
2. A two-minutes timer.
3. The participant's extraction and payoff in the previous round
4. The total group extraction in the previous round
5. The participant's extraction for the round (this is where the default was shown)
6. An interactive sandbox where they can simulate their and others' payoffs and a table with all the possible group extractions, and the ECU's produced.

Additionally, the participants knew that they are interacting in groups of 4 which remain fixed for the duration of the CPRD task, that the game is played over 10 rounds, and that there is a maximum of 2 minutes to make a decision. Participants were instructed to make their decisions within this time, otherwise they will be considered as dropouts from the experiment. Also, after every round, the participants know how much the other three participants extracted in the previous round, and they are notified if a co-player dropped out of the experiment, in which case they continued playing in a smaller group, and got paid. Dropouts, totalling 21 instances (2.8% of the total participants) due to *Timeout* or *Lost focus* (see S1 Table), led to the exclusion of data from those completing the ten rounds. S2 Table indicates

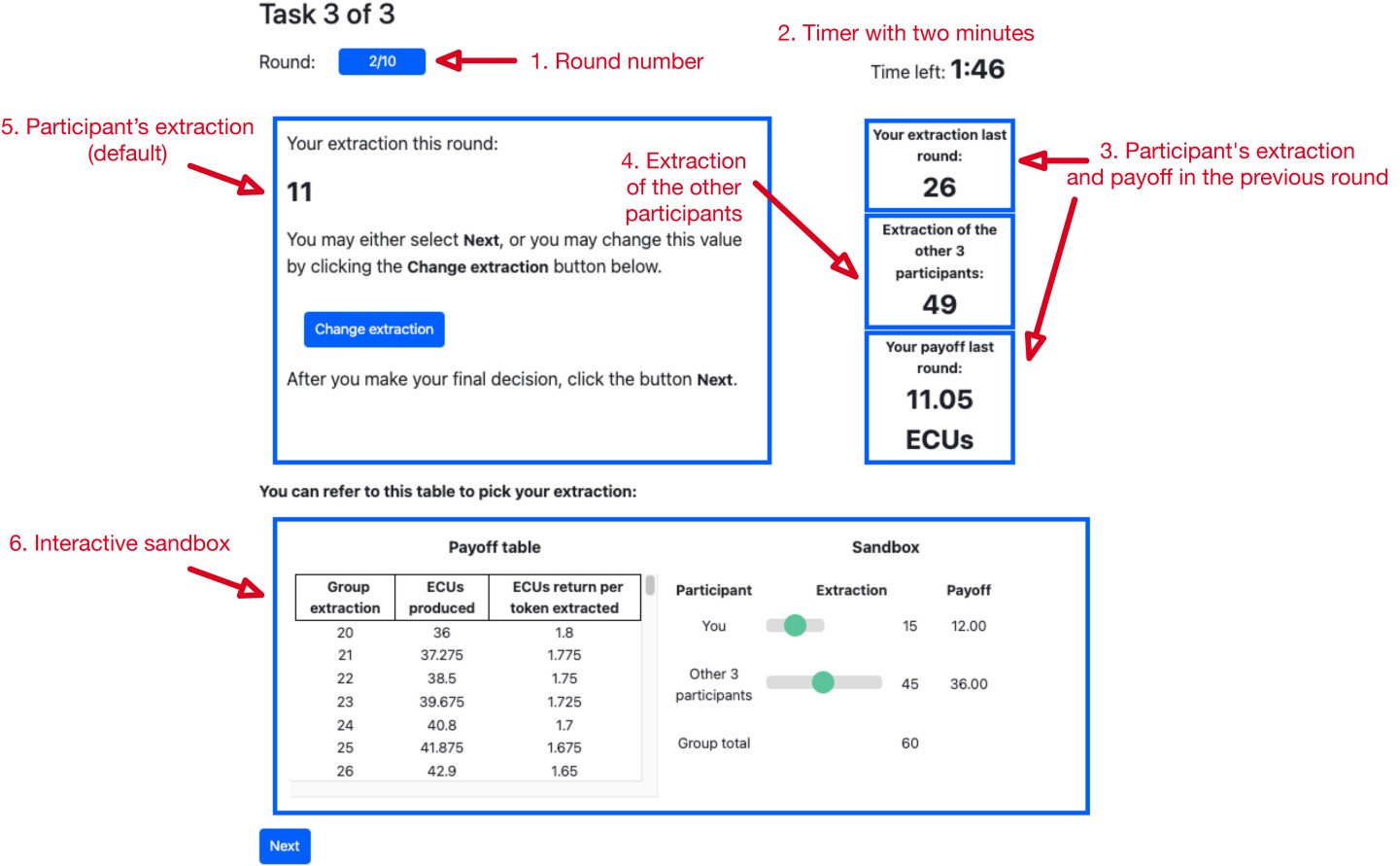

**Fig 2. Screenshot of the third task.** In this task, participants had to choose their desired extraction for ten rounds. At the top, the platform showed the round number and the time left to make their decision. The participants had their previous extraction and the extraction of the other members of the group, and their payoff, as shown on the left side of the figure. At the bottom, participants had a sandbox at their disposal to calculate their potential earnings depending on theirs and others' extraction.

that previous tasks' results, demographic information, do not predict individual drop-out, mitigating selective attrition concerns. In the simplified regression presented in Supplementary S3 Table, we fitted a reduced logit regression including only task-related variables (SVO score and gamble choice). We found no significant association between SVO score or gamble choice and dropout.

All participants had to complete a comprehension test for task 3, in which they were given five attempts to get the five questions correct. If they could not complete this test, they were paid for the earnings in the previous two tasks, but they were excluded from playing the CPRD game. All the instructions and captions can be found in the Supporting documents.

**2.1.2 Experimental treatments.** The treatments consist in the introduction of default extractions. Every participant in the group was assigned to the same treatment. We implemented defaults by pre-setting a token extraction value. Participants had the possibility to override this value. We also include a Control treatment with no default values, i.e., participants always had to choose their own extraction. Both in the Control treatment and the last five rounds of the Pro-social and Self-serving treatments, this was enforced by making participants choose a value between 1 and 30, before proceeding to the next round. In this case, what they saw in the interface then was a drop-down menu, with two dashes (–) meaning that they

had to pick an amount. We compare the effect of the manipulation done in the two default value treatments with this control.

The values were chosen based on the social optimum and an exploitative extraction:

- Social optimum: $x_f = \frac{a}{2bn}$ where n is the group size.
- Exploitative extraction: $x_h = \frac{a}{bn}$

Given the parameters we chose ($a$ = 2.3 and $b$ = 0.025, see previous section), $x_f$ = 11.5 and $x_h$ = 23. We rounded $x_f$ to 11 because participants can only pick an integer from the user interface. If in any given round, $x_f$ is collectively chosen, i.e. the group extraction is 44, the resource will yield its maximum. This also means that participants can get more if they stick with this extraction. However, players will be enticed to pick a higher extraction to get a higher share of the group extraction. When the group extraction reaches 92, or $x_h$ is chosen by everyone, the group and individual payoffs are zero. The Nash equilibrium in this game is defined as $x^* = \frac{a}{b(n+1)}$, in which, if all players are self-interested and rational, they will pick this reaching a suboptimal equilibrium [62,63], as shown in S1 Fig in the Supporting Information. We chose $x_f$ and $x_h$ to test the default effect with the social optimum extraction of the CPRD, but also to show how a socially detrimental value such as $x_h$ is able to anchor individuals' extraction higher than the Nash Equilibrium.

The group payoff depending on the total of extraction can be found in S1 Fig in the Supporting Information. The coloured dashed lines in the figure indicate the amount that is given back to the players if all members of a group selected a certain default value.

Participants are subjected to one of the 3 treatments:

- **Control - No default value,** $n$ = 100: in this control treatment, participants play the CPRD without default extraction proposed for 10 rounds.
- **Pro-social treatment (Pro-social),** $n$ = 156: in this treatment, participants are shown a default value representing the social optimum value $x_f$ with a label that reads: *Your extraction this round*: 11. At each round, each participant has the option to override this value, and choose another extraction for themselves. The wording is neutral, and no notion of "fairness" or "fair extraction" is communicated to the participants.
- **Self-serving treatment (Self-serving),** $n$ = 156: in this treatment, participants are shown a default value representing the exploitative value $x_h$ with a label that reads: *Your extraction this round*: 23. At each round, each participant has the option to override this value, and choose another extraction for themselves. The wording is neutral, and no notion of "exploitative" or "selfish extraction" is communicated to the participants.

We designed the experiment to have a higher sample size for the treatments than the control, because we hypothesised that there would be more heterogeneous effects in the former (in function of the individuals' social preferences and the default, see Sect 2.2). Nevertheless, we ensured that the sample size in the control treatment is sufficient to achieve the statistical power required.

To change the default extraction, participants have to click a button below the extraction and pick a value from a drop-down menu. To measure persistence, in both treatments (Pro-social and Self-serving) participants are subjected to the default value manipulation for 5 rounds, while in the subsequent 5 rounds, no default option is presented, i.e., participants have to manually pick the extraction they desire. Our experimental manipulation is "between-subject": participants could not take part in more than one treatment.

The currency used in this experiment is ECU (Experimental Currency Unit). Each ECU is converted to U.K. Pounds (£) with an exchange rate of or 100 ECUs = £1, or 1 ECU = £0.01. Participants were recruited using Prolific (www.prolific.com). In Prolific, we recruited participants who indicated they were fluent in English, had a high approval rate (+99%) and had more than 20 submissions on the site. These filters were applied to filter out possible dropouts. Participants received £3 as a fixed participation fee plus £6 bonus depending on the decisions they made in the game. The average total earnings were £5.05 for a duration of roughly 30 minutes.

We excluded participants who did not sign the Informed Consent Form, dropped out, failed the comprehension test, or did not act within the time limit for each round. Participants with group drop-outs are also excluded, even if they finished the experiment themselves. Additionally, participants who were excluded or dropped out from the experiment were only paid for the tasks they completed. This design choice aimed at motivating participants to finish all tasks, while rightfully paying them for their time and for the completed tasks. Therefore, the experimental data used in this paper are from participants in groups of four who completed all ten rounds, passed the comprehension test, and successfully submitted their work on Prolific.

The raw experimental data, instructions with screenshots included, is attached to the Supporting Information. All experiments described in this paper followed the guidelines and regulations of data protection and experiments with human participants, and they were approved by the Ethical Commission for Human Sciences at the Vrije Universiteit Brussel (VUB) in Brussels, Belgium (ECHW_361.02). All procedures performed in studies involving human participants were in accordance with the ethical standards of the institutional and/or national research committee and with the 1964 Helsinki Declaration and its later amendments or comparable ethical standards. Also, all participants who took part in the experiment signed an Informed Consent form for the use of the data collected in the experiment, including decisions and background information, including gender and age. Without signing this form, they could not proceed with the experiment.

## 2.2 Personal preferences

For the first task of the experiment, we assessed participants' social preferences using the Social Value Orientation measure (SVO) [64] and the Slider measure by Murphy et al. (2011) [50,61] with a modification to include incentives (see S2 Fig in the Supporting Information for a sample of the decisions shown to participants). Participants made six allocation decisions. Participants were informed that they would be matched with another participant at the end of the experiment. In each match, the allocation decisions of one of the participants would be played out (described as Group A), with the other one being paid as a receiver (described as Group B). When decisions are made, one of the six choices was randomly selected as the final allocation between a participant from Group A and another from Group B. Participants in Group B will receive an allocation determined by a member of Group A. Their resulting SVO were represented as an angle from the origin, where a higher angle means greater cooperativeness, and lower angles indicated individualistic or competitive allocations. For an easier analysis, we grouped subjects with SVO angles < 22.45° as "Individualistic+Competitive" and SVO >= 22.45° as "Cooperative+Altruistic" (as done in other works [58,65]) in some sections of the document instead of using the continuous angle.

In the second task, we implemented a risk-attitude elicitation task as done by Eckel and Grossman [52], with the values from Dave et al. (2010) [51] Participants have to decide between six different gambles, from the least risky to the most risky, of an event happening

with 50% chance. A coin was flipped to pick which of the events will pay them and that was their payoff for their task. The resulting measure is a discrete range from 1 to 6 where 1 is the safest and 6 is the riskiest (see S3 Fig in the Supporting Information for a sample of the options shown to participants).

We acknowledge the possibility that eliciting SVO and risk attitudes prior to the main CPRD task could influence participants' subsequent behaviour by priming social or risk-related considerations. However, we opted to measure these preferences beforehand to avoid potential contamination from the CPRD game itself, which involves social and strategic interactions that could bias SVO if measured afterwards. Since the order of tasks was constant across all experimental conditions, any potential priming effects would be consistent across groups and thus unlikely to confound treatment comparisons. For this reason, any potential spillover would not confound the treatment comparisons central to our design.

## 2.3 Mixed models

We fitted a generalized additive (mixed) regression model (GAM) for the panel data using the `mgcv` *R* package for *GAM* estimation [66,67]. We used this to test the interactions of the variables over time and by treatment. The variables we want to study to understand the extraction behaviour were:

- Treatment (or the default value presented, if any)
- Round number
- SVO score in task 1
- Gamble choice in task 2
- Player's extraction in the last round
- Others' extraction in the last round

Our aim is to test the hypothesis that the default value presented in the Pro-social and Self-serving treatments will have different effects in function of the social and risk preferences of different subjects. Specifically, depending on their SVO score in task 1 and gambling choice in task 2. The reason behind this is that we hypothesised, for example, that cooperative individuals will react to a high default differently compared with an individualistic person. Also, we assumed random intercepts for each individual, and each group, to account for individual-level variability in our subject pool and group-level variation not captured by the fixed effects. Hence, our `lme4` formula looked like this:

$$x_{i,r} \sim T + T \times \text{SVO} + T \times r + T \times \text{gamble} + x_{i,r-1} + X_{-i,r-1} + (1|i) + (1|G) \tag{2}$$

Where $x_{i,r}$ is the extraction of the player $i$ in round $r$ where they were in treatment $T$, also $X_{-i,r-1}$ represents the extraction of the three other group members, and the term $(1|i)$ in the equation represents the random effects for player variability. The last term, $(1|G)$ accounts for random effects at the group level to control for unobserved variability between experimental groups.

We also show the difference in extraction between two levels of a categorical variable, in this case we use this model to show the time window where the extraction between two treatments is significantly different according to the model. A significant time window is given by the estimated difference in extraction, where the confidence intervals of the prediction do not include zero. This is useful to show the evolution of the default manipulation and how it affects individuals with different SVO scores.

# 3 Results

## 3.1 While defaults influence participants, the effect is asymmetrical

In the experiment flow (see Fig 1), participants were randomly assigned to one of three treatments in the third stage (see Methods and Table 1 for details of each treatment): In the *Self-serving* treatment ($n$ = 156 participants, 39 groups), the default extraction shown to participants in the first five rounds corresponds to an individually beneficial extraction (i.e. extraction value $x$ = 23). This default could potentially yield a high individual payoff, but if collectively chosen, results in all participants obtaining zero payoff. In the *Pro-social* treatment ($n$ = 156 participants, 39 groups), the default was set to an extraction value beneficial for resource and society (i.e. extraction value $x$ = 11). This choice is a mutually optimal one, which, if selected by everyone, leads to an individual payoff of 132 ECU and maximises collective surplus. Lastly, in the *Control* treatment ($n$ = 100 participants, 25 groups), no default value was shown, and thus each participant had to select how much to consume in each round. In CPRD theory, rational, self-interested participants extract the symmetric Nash equilibrium ($x$ = 18, depicted as the blue dashed line in Fig 3), leading to resource overexploitation and a suboptimal equilibrium [28].

Moreover, we considered the impact of dropouts during the experiment (see the output of this regression in S10 Table in the Supporting Information section). First, we tested whether dropout rates varied significantly across treatment groups. The chi-square test yielded a non-significant result, $\chi^2(2)$ = 0.86, $p$ = 0.65, indicating that dropout rates did not differ systematically between treatments. Second, we replicated our analysis including dropouts in the regression model, and the overall findings remained qualitatively unchanged. This suggests that our results are not driven by dropout.

In the first five rounds of the CPRD (see Fig 1), where participants of the Self-serving and Pro-social treatments were shown a default value, the mean extraction did not start exactly from the default option but was effectively nudged to more or less extraction levels depending on the default value presented to them, as shown in Fig 3A. The pink markers in Fig 3A show that participants in the Self-serving treatment extracted more ($\bar{x}$ = 18.78, 95% CI = [18.32,19.23]) in the first five rounds than those in the control ($\bar{x}$ = 15.93, 95% CI = [15.31,16.54]) and Pro-social ($\bar{x}$ = 15.12, 95% CI = [14.71,15.53]) treatments. An analysis of variance (ANOVA) shows for the first five rounds that the average extraction over time is significantly different between treatments ($F(2)$ = 31.82 Control ($n$ = 100), Pro-social($n$ = 156), Self-serving ($n$ = 156) , p-value =< 0.01). The Self-serving default nudged participants towards the Nash equilibrium, particularly in rounds 3 to 5, with a mean extraction of $\bar{x}$ = 18.69 and a 95% CI of [18.08,19.29], not significantly different from the Nash equilibrium (Wilcoxon signed-rank test, V = 6997, ($n$ = 156) p-value = 0.06).

As long as the default value was shown, the participants in the Self-serving treatment extracted significantly more than the in control treatment ($F(1)$ = 25.80, Pro-social ($n$ = 156), Control = ($n$ = 100), p-value =< 0.01). For the Pro-social treatment, this difference extended

**Table 1. Descriptive statistics of the experimental treatments.**

| Treatment | No. Participants | No. Groups | Default | Extraction (all rounds) | |
|---|---|---|---|---|---|
| | | | | Mean | 95% CI |
| Pro-social | 156 | 39 | 11 | 15.72 | 15.42, 16.02 |
| Self-serving | 156 | 39 | 23 | 17.84 | 17.50, 18.17 |
| Control | 100 | 25 | – | 16.32 | 15.89, 16.76 |

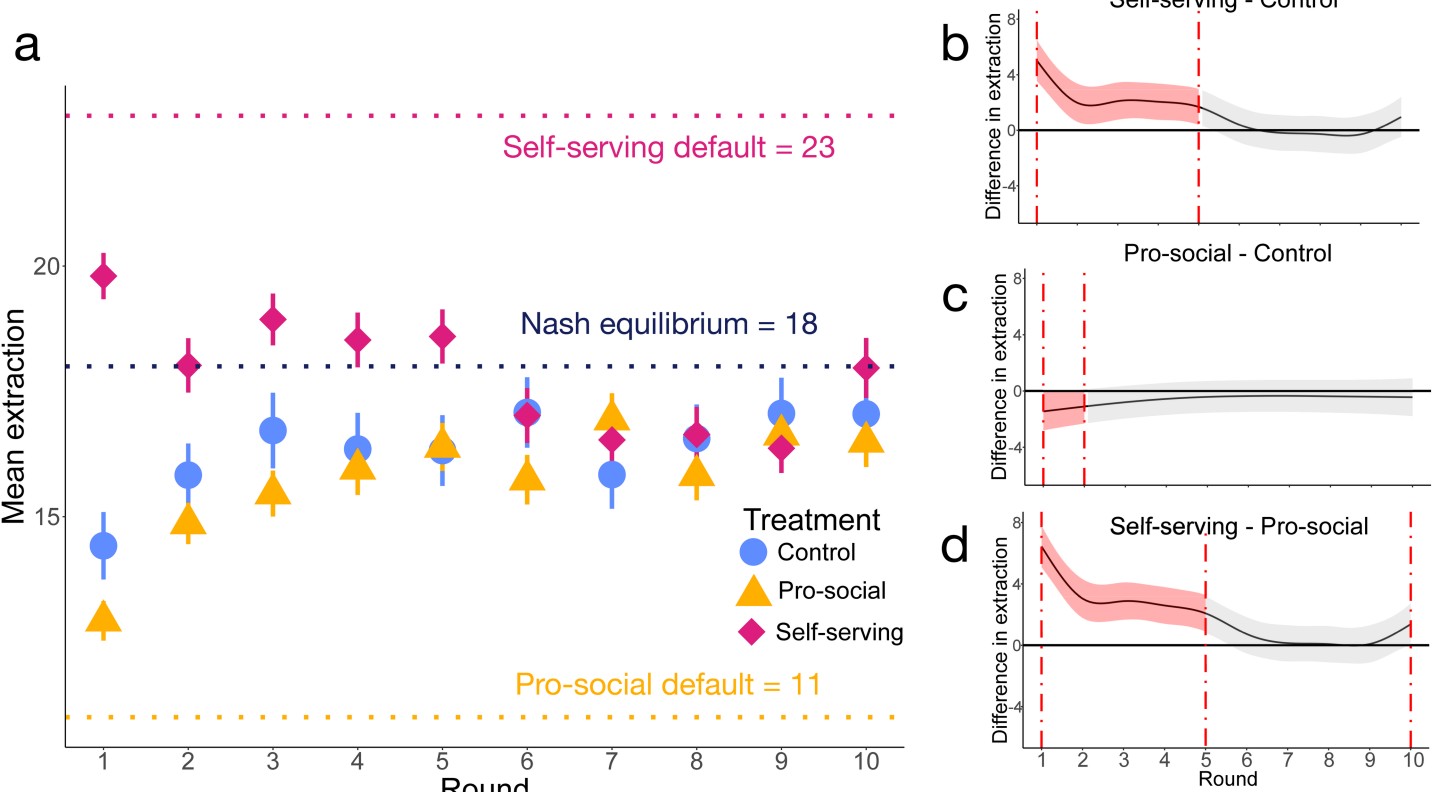

**Fig 3. Mean extraction per treatment. a)** Mean extraction by round and by treatment: Pro-social $n$ = 156, Self-serving $n$ = 156 and Control $n$ = 100. The blue dotted line represents the Nash Equilibrium of the game, while the dotted lines on both extremes represent the default values presented to participants. Vertical lines represent 95% confidence interval. **(Panels b, c and d)** Estimated difference of mean extraction between treatments the red is the estimated difference, as given by the mixed-effect model, while the vertical lines represent the round where this significant difference can be found. The colour shaded areas represent the 95% confidence interval from the model.

until round 3, but thereafter the difference was no longer significant ($F(1)$ = 2.30, Pro-social ($n$ = 156), Control ($n$ = 100), p-value = 0.13).

To visualise these differences, Fig 3B, C and D show the estimated differences in the extraction over rounds using a mixed-effects model (see Methods for more details on this model). This model can identify the time periods where the extraction is significantly different from zero. The area in pink between the red vertical lines represents the area where differences between treatments are significant at the 5% level. The fit between the model and the experimental data can be seen in S8 Fig in the Supporting Information.

Fig 3B shows that the participants extracted less (below the zero line) in the control treatment than in the Self-serving treatment in the first five rounds, i.e. where the default value was shown. Fig 3C shows how participants extracted more (above the zero line) in rounds 1 and 2, and then the mean extraction remained the same on average in the remaining rounds. Fig 3D shows the mean difference in extraction between the Pro-social and Self-serving treatments.

In this regard, the default effect showed an asymmetrical duration: the effect of showing a selfish default value lasted for longer, compared to showing a pro-social default value. Indeed, if the effect was symmetrical, participants in the Pro-social treatment would extract significantly less (compared to showing no default) for the same duration as they did in the Self-serving treatment. This difference is reflected in the absolute slopes of the regression model

(see Sect 2.3 for more details), which indicate the rate of change in extraction behaviour. In the early rounds, participants in the Pro-social treatment extracted significantly less, as indicated by the steeper absolute slope. (see S9 in the Supporting Information for more details). The selfish default was more effective to anchor participants to over-extract than the pro-social default was to nudge participants toward sustainable levels of extraction.

## 3.2 The effect of all defaults disappear as soon as they are lifted

After round 6, where participants must choose an extraction without a default as in the Control treatment, the participants in the three treatments had similar mean extractions, as shown in the gray shaded areas of Figs 3B, C and D. Actually, in the last five rounds, the mean extractions of the Self-serving ($\bar{x}$ = 16.91, 95% CI = [16.42,17.39]) and Pro-social ($\bar{x}$ = 16.33, 95% CI = [15.88,16.77]) treatments are very similar to the mean extraction in the Control treatment for all rounds ($\bar{x}$ = 16.72, 95% CI = [16.12,17.32]). An analysis of variance (ANOVA) further confirms this, as extractions are not significantly different between the three treatments after round 6 ($F(2)$ = 0.64, Control ($n$ = 100), Pro-social ($n$ = 156), Self-serving ($n$ = 156) p-value = 0.53). Moreover, the presence of an exploitative default was a significant factor whilst present, (see S7 Table for more information). Note that the mean extraction in all treatments is closer to the symmetric Nash equilibrium (18 tokens, shown in the blue dotted line in Fig 3A) than to the default values provided in the treatments. Therefore, the effect of defaults fades away as soon as the defaults are lifted, which contradicts the "stickiness" hypothesis.

## 3.3 Cooperatives can be nudged to extract more and selfish participants to extract less

We hypothesised that extractions are affected not only by the defaults but also by personal preferences (which were measured in Task 1 and 2) and that there could be interesting interactions between the default value and both SVO and Risk preferences.

The results of Tasks 1 and 2 allows us to study the personal preferences of the participants and to link them with the behaviour in the third task. S4 Fig in the Supporting Information presents the distributions of SVO, measuring social preferences, and gambles choices, measuring risk preferences. Most participants across all treatments fall under the "cooperative" spectrum ($n$ = 275, ≈ 67%, 22.45° < $SVO°$ < 57.15°) and another share under the "individualistic" ($n$ = 132, ≈ 32%, 12.04° < $SVO°$ < 22.45°) trait in this task. Three participants can be classified as "altruistic" (≈ 0.7%, $SVO°$ > 57.15°) and two as "competitive" (≈ 0.4%, $SVO°$ < 12.04°). Moreover, 283 participants (69%) picked the three least risky gambles (the lower, the safest) in the Risk Assessment task. S4 and S5 Figs in the Supporting Information show the distribution of extraction by SVO scores and gamble choices.

To link the individual data from the first two tasks with the data of Task 3, we fitted a mixed-effects model (see Methods section for the details on the fitting). The results of the regression model are detailed in S8 Table in the Supporting Information, there is a significant interaction term between SVO score and the treatments, which means that the extraction participants made depended on both the default value presented and their SVO score. Moreover, we accounted for group non-independence in our mixed-effects model by including random intercepts at the group level (see S9 Table), also for those groups with fewer than four members, see S10 Table. Lastly, as a robustness check, we report the cluster-robust standard errors at the group level (see S11 Table, all in the Supplementary Information). We find the main results remain valid even with these robustness checks, as the new random effect for the group-level is not significant.

The model also reveals a difference in extraction based on SVO and CPRD round, which we visualise in the form of a heatmap in Fig 4. This figure shows the extraction difference between two treatments, and where a significant difference was found. In panel **A**, the difference shown is negative, meaning that the participants in the Self-serving treatment extracted more than in the Control treatment, for the first five rounds and mostly on the upper bounds of the SVO spectrum, i.e. where Cooperative and Altruistic preferences reside. This means that the Self-serving default presented led cooperative and altruistic individuals to extract more on average ($F(1) = 20.02$ Control ($n = 73$), Self-serving ($n = 102$), p-value =< 0.01) than what they would do in the Control, as also shown in Fig 4C. In panel **B**, where the Pro-social treatment is compared to the Control, this difference is mostly in the first three rounds for the lower bounds of the SVO spectrum, i.e. Individualistic and Competitive participants. Similar to the previous finding, the Pro-social default makes selfish participants extract less, with respect to the Control treatment ($F(1) = 5.02$ Control ($n = 25$), Pro-social ($n = 56$), p-value = 0.03). This difference can be seen in Fig 4D where the highlighted markers show the mean

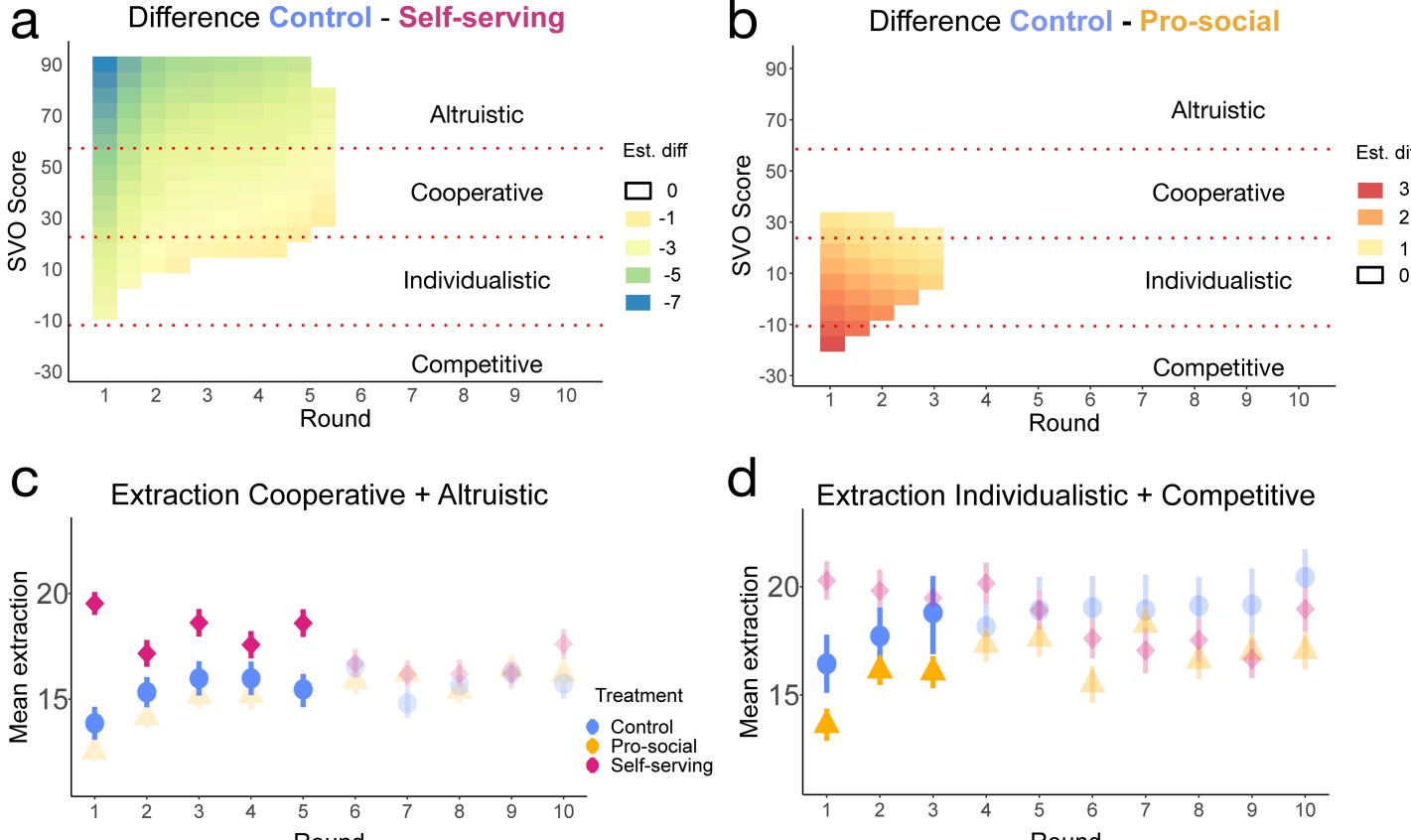

**Fig 4. Difference in extraction depending on participants' SVO over time. Panels a and b:** Estimated difference given by the mixed-effects model, where non-white parts show a significant difference between two treatments. The horizontal lines show the SVO categories for reference and the colours the difference in extraction. Negative (or positive) differences show a larger (or smaller) extraction than the results in the Control treatment. **Panels c and d:** Mean extraction over time given by the experimental data, where the most pro-social categories of SVO (cooperative and altruist) are grouped in panel **c** and the most selfish categories of SVO (individualistic and competitive) are grouped in panel **d**. The highlighted points in both panels are the ones given by the model in panels a and b. Vertical bars show the 95% confidence interval.

extraction in treatments Control and Pro-social. Lastly, the Pro-social default did make cooperative and altruistic individuals extract less $\bar{x}$ = 14.54, 95% CI = [14.0400,15.04] than in the Control $\bar{x}$ = 15.37, 95% CI = [14.64,16.01] (KS-statistic = 0.22, Control ($n$ = 73), Pro-social ($n$ = 100), p-value = 0.01).

Regarding risk preferences, we classified the subjects who picked gambles 1 to 3 as "risk-averse" and those who picked gambles 4 to 6 as "risk-seekers" (see Methods for details about the gambles). We found that in the Self-serving treatment, risk-seekers extracted less on average ($\bar{x}$ = 17.88, 95% CI = [16.98,18.78]) than risk-averse subjects ($\bar{x}$ = 19.16, 95% CI = [18.63,19.69]), however, this difference is not significant (KS-statistic = 0.19 Risk-averse ($n$ = 109) Risk-seeking ($n$ = 47), p-value = 0.08). Conversely, risk-seeking subjects extracted more in the Pro-social treatment ($\bar{x}$ = 16.39, 95% CI = [15.57,17.21]) than risk-averse ($\bar{x}$ = 14.57, 95% CI = [14.11,15.03]), and the difference between these two means is significant (KS Statistic = 0.21 Risk-averse ($n$ = 109) Risk-seeking ($n$ = 47), p-value = 0.04), see Fig S6. With this finding, we could observe how the risk-averse participants conform with the proposed default more often, resulting in more average extraction in the Self-serving treatment and less extraction in the Pro-social, with respect to their risk-seeking counterparts.

## 4 Discussion

Our experimental results show that default extraction levels have a significant impact over overall extraction levels in a CPRD. Notably, we find that the effect of two opposite default options have asymmetrical effects: although both tested defaults (Self-serving and Pro-social) had an effect on the first round of the experiment, their persistence differed significantly. This decay mirrors Chaudhuri and Paichayontvijit's (2017) finding that cooperation dropped after removing exhortative messages to cooperate [38]. Participants in the Self-serving treatment extracted more on average over five rounds, while those in the Pro-social group extracted less only in the initial two rounds. Thus, in both treatments, while the effect does not persist for the entirety of the CPRD task, the effect remains significant for more than one round, a result that agrees with previous research on the spillover effect of defaults [40,41].

It is important to highlight that in the CPRD, as long as the group extraction is below the Nash Equilibrium (NE) participants have a strategic incentive to increase their extraction level. For this reason, coordinating the decisions of a group in the CPRD towards the socially optimum level of extraction is a challenge. In this context, our experiment shows that a Pro-social default has the least negative effect on cooperation and offers a potentially cost-effective solution to drive such social behaviour. Moreover, although on average the effect on the decrease of overall extractions only persisted for 2 rounds, we find that they had a strong impact on individualistic and competitive individuals for at least 3 rounds. Nevertheless, this general trend, intrinsic to the CPRD, of converging towards a level of extraction close to symmetrical NE [3] can be observed in both treatments, which converge to an extraction value that is, on average, closer to the control treatment, particularly, after the default intervention is lifted (after round 5). This convergence resembles Fehr and Gächter's (2000) collapse to Nash-like outcomes when punishments are removed [39].

In this line, one of the main goals of this work relied on identifying potentially heterogeneous effects of defaults over the experimental population in function of social preferences. As discussed in Sect 3.3, selfish individuals reduced their extraction for three rounds when faced with a pro-social default, while cooperative participants increased their extraction in response to a selfish default while this intervention was present. Given these results, default values likely act as temporary focal points, establishing a new social norm [68,69].

Moreover, while defaults can nudge behaviour in the short term, other mechanisms might help regulate extractions over time. Our results show that the pro-social default vanishes quicker compared to the self-serving default, likely because most individuals are "imperfect conditional cooperators," as demonstrated by Fischbacher et al. (2010), who found that contributions in public goods games decline as people only partially match others' contributions. Over time, even groups not fully driven by income maximisation (like cooperatives) adopt selfish behaviours, reducing overall cooperation [16]. This might explain the higher average extraction in the Self-serving default and its slower convergence to the NE. This gradual shift explains why the effects of defaults, whether pro-social or exploitative, are often temporary and fade as participants adjust their behaviour based on evolving group dynamics.

Our work highlights the potentially strong negative impact of establishing selfish focal points as defaults. While our experimental setting clearly distinguishes between pro-social and selfish defaults, this may not be as apparent in other contexts. Our findings suggest that defaults prioritising individual comfort are not only more readily adopted than socially beneficial options but may also influence the behaviour of those inclined toward sustainable practices, such as cooperatives. Thus, in systems where a default value has to be enforced by design, setting up the "best practices" as defaults, may be an effective strategy to mitigate over-consumption [18,21]. On the other hand, setting the "wrong" default can have even stronger (negative) consequences, and spoil the decisions of previously pro-social participants [70]. While good defaults can nudge otherwise selfish people, bad defaults have the potential to do the inverse to a greater extent.

Overall, default options are a simple and cost-effective tool to support consumer decision-making in problems where the effort required to collect enough information to make optimal collective decisions is high. In this sense, they have the potential of being powerful tools for policy-makers in social problems that require resource management and environmental conservation [15,71], laws aimed at mitigating dark patterns in digital services and markets [72], enhancing data privacy [73], and improving products' environmental performance [74] through the correct use of defaults can prove beneficial. Additionally, the heterogeneous effects we find in this work indicate that policies should be tailored to the targeted population and context. Finally, it is important that defaults align with individual and societal welfare, emphasising ethical considerations and autonomy preservation of the individuals.

## 5 Limitations

Our experiment uses an abstract game as a proxy for real-world resource consumption. To improve the generalisation of our results, we avoided using loaded language or contextual instructions. While this allows for a greater control and interpretability of participants' behaviour, this also sidesteps important dimensions that are present in many (politicised) real-world settings, such as emotional involvement in the preservation of the environment, notions of fairness or the idea of global responsibility. Thus, future use of our findings may require further testing to account for contextual specificity of the desired application setting.

Moreover, since in this experiment we did not aim to collective a representative population sample over all demographic subgroups, such as age, gender, or nationality, we did not take these factors into account in our analysis. While we provide general demographic information (see S4, S5 and S6 Tables in the Supporting Information), we did not investigate how these factors might influence our results. Future research should address these issues by collecting data from more specialised groups and exploring potential differences across demographic subgroups.

Furthermore, our study did not account for the long-term behavioural changes of participants, which suggests the need for longitudinal studies to observe the durability of the observed behaviours in the long term.

## 6 Conclusions

Our experiment reveals that exposing participants to a default extraction value in a CPRD significantly affects their extraction patterns. Specifically, we found that participants in the Self-serving treatment extracted significantly more resources than those not exposed to any default values. Additionally, its impact varies based on the participant's SVO. Cooperative individuals exhibited selfish behaviour by extracting more when faced with a Self-serving default, while, notably, selfish individuals extracted less when confronted with a Pro-social default. Furthermore, the default effect is non-persistent, and its impact on participants' decisions diminishes rapidly once the default is removed.

Our findings have substantial implications for designing systems and policies aimed at the sustainable management of common resources. They highlight the importance of accounting for individual heterogeneity and demonstrate the potential of simple, low-cost interventions like defaults in nudging groups towards more socially beneficial outcomes.

## Supporting information

Supplementary Information: document with extra tables, figures and analysis.

**S1 Fig Group payoff as a result of a given group extraction.**
(TIF)

**S2 Fig Sample of three out of six decisions participants had to make in the first task.**
(TIF)

**S3 Fig Sample of three out of six options participants had for the second task.**
(TIF)

**S4 Fig Left: Distribution of participants according to their SVO Score (task 1).**
(TIF)

**S5 Fig A: relationship between participant's SVO score in task 1 and their mean extraction in all rounds.** B: Mean extraction of the participants according to their gamble choices in Task 2 of the experiment.
(TIF)

**S6 Fig Gamble choice and extracting the default option.**
(TIF)

**S7 Fig Post hoc power calculation based on the obtained sample using G*Power.**
(TIF)

**S8 Fig Mean extraction by treatment, with the fit given by the Mixed-effects regression model.**
(TIF)

**S9 Fig Absolute values of the slopes for each treatment of the regression of the mixed model.**
(TIF)

## Author contributions

**Conceptualization:** Eladio Montero-Porras, Rémi Suchon, Elias Fernández Domingos.

**Data curation:** Eladio Montero-Porras.

**Formal analysis:** Eladio Montero-Porras, Rémi Suchon, Elias Fernández Domingos.

**Funding acquisition:** Tom Lenaerts.

**Investigation:** Eladio Montero-Porras, Rémi Suchon.

**Methodology:** Eladio Montero-Porras, Rémi Suchon, Elias Fernández Domingos.

**Project administration:** Eladio Montero-Porras.

**Resources:** Eladio Montero-Porras, Tom Lenaerts.

**Software:** Eladio Montero-Porras.

**Supervision:** Rémi Suchon, Tom Lenaerts, Elias Fernández Domingos.

**Validation:** Eladio Montero-Porras, Elias Fernández Domingos.

**Visualization:** Eladio Montero-Porras.

**Writing – original draft:** Eladio Montero-Porras.

**Writing – review & editing:** Eladio Montero-Porras, Rémi Suchon, Tom Lenaerts, Elias Fernández Domingos.

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
