## [Decision Letter · Decision Letter 0]

25 Mar 2025

PONE-D-25-04274From self-interest to collective action: The role of defaults in governing common resourcesPLOS ONE

Dear Dr. Montero-Porras,

Thank you for submitting your manuscript to PLOS ONE. After careful consideration, we feel that it has merit but does not fully meet PLOS ONE’s publication criteria as it currently stands. Therefore, we invite you to submit a revised version of the manuscript that addresses the points raised during the review process.

Three reviewers provided valuable comments on your manuscript. 

I also reviewed your manuscript throughout and confirmed that it was well-written and there seemed to be no major problems for methodology and results. 

I also agree with their comments about the necessity of improvement for your analysis to adjust for a random effect of groups. 

Please check the reviewer's comments and revise your manuscript according to the comments with point-by-point responses.

We look forward to receiving your revised manuscript.

Kind regards,

Yutaka Horita

Academic Editor

PLOS ONE

2. Please note that your Data Availability Statement is currently missing [the repository name and /accession number of each dataset OR a direct link to access each database]. If your manuscript is accepted for publication, you will be asked to provide these details on a very short timeline. We therefore suggest that you provide this information now, though we will not hold up the peer review process if you are unable.

Additional Editor Comments (if provided):

Reviewers' comments:

Reviewer's Responses to Questions

**Comments to the Author**

1. Is the manuscript technically sound, and do the data support the conclusions?

Reviewer #1: Yes

Reviewer #2: Yes

Reviewer #3: Partly

2. Has the statistical analysis been performed appropriately and rigorously? 

Reviewer #1: Yes

Reviewer #2: Yes

Reviewer #3: Yes

3. Have the authors made all data underlying the findings in their manuscript fully available?

Reviewer #1: Yes

Reviewer #2: No

Reviewer #3: Yes

4. Is the manuscript presented in an intelligible fashion and written in standard English?

Reviewer #1: Yes

Reviewer #2: Yes

Reviewer #3: Yes

5. Review Comments to the Author

Reviewer #1: In this paper, the authors tested the effects of prosocial and self-serving defaults on extraction levels in a common-pool resource dilemma. The results of an online economic game showed that prosocial defaults caused people to extract less from the common pool and self-serving defaults caused people to extract more. However, these effects were short-lived, disappearing entirely when defaults were removed halfway through the game. The effects were also moderated by Social Value Orientation and risk preferences: for example, the authors found that prosocial defaults were more effective for individualists and risk-averse participants.

The study was described very clearly and the results were easy to follow. I really liked the statistical approach taken in the paper – more studies with repeated games should use GAMs to track changes across rounds! I also assessed the pre-registration, data, and analysis code as part of my review. The pre-registration was comprehensive and clearly laid out the key hypotheses. I was able to reproduce the statistical results of the paper with the data and analysis code (after some changes to file paths in the R file).

That said, I had a few concerns about the paper, mainly related to the discussion of prior literature, the statistical analyses, and the treatment of dropouts.

First, I felt that the discussion of prior literature was scant in places, making it difficult to assess how much of a contribution this paper is making. For example, in the Introduction section, the authors could have discussed how previous psychological work has already studied how the effects of defaults vary depending on individual differences in anxiety and avoidance (Zucchelli et al. 2024), cognitive effort (Ortmann et al. 2023), and mood and personality (Manganari et al. 2022). In the Discussion section, it might also be worth discussing how the results are in line with other studies in behavioural economics, particularly studies showing that the effects of other interventions in repeated games often decay over time (e.g., Chaudhuri & Paichayontvijit, 2017) and go away entirely when removed (e.g., Fehr & Gächter, 2000). Regarding generalising the results to real-world settings, the authors could discuss the use of green defaults in airline carbon offsetting (Berger et al. 2022) and speculate as to whether such interventions might be more effective for more individualist and risk-averse customers. I am aware that the literature on defaults is very large, and this is not a review paper! But I think some extra discussion in places could better couch the current paper and its findings in the existing literature.

Second, I had a few concerns with the analysis approach. While I liked the use of GAMs, the authors should consider including random intercepts for groups in the models in addition to the random intercepts for participants. The nested nature of the experiment means that there will be dependencies within groups. This group-level variation may not be fully captured by the “previous group extraction” fixed effect in the model. Do the results hold when this group-level variation is accounted for with additional random intercepts?

Third, there is the perennial issue of dropouts in online experiments. If I’m understanding correctly, the authors’ pre-registered approach was to exclude any participants who dropped out and any groups containing dropouts. But this is problematic from a causal inference perspective, especially if dropouts are more common in one treatment group compared to others – if this is the case, it becomes difficult to know whether differences between experimental conditions are due to the treatment itself or due to differences in attentiveness and characteristics of participants who drop out. The authors could remedy this in two ways: (1) by analysing whether dropout rates vary systematically across treatment groups, and (2) by running an “intention-to-treat” analysis (McCoy, 2017) that fits the same models but retains observations from groups that contain dropouts. Since there are only a few dropouts, the results will likely not differ all that much, but this analysis would show that the main causal conclusions are unbiased.

Finally, here are some minor additional points to consider:

• Some visual examples of “free-riding” and “cooperation” in the economic game might be helpful for readers, especially those outside of behavioural economics.

• I am assuming that the treatment is at the group-level, such that all participants within a group experience the same treatment. If this is the case, it should be clearly stated in the Methods section.

• Figures 1b and 1c are slightly confusing because the splines go in the opposite direction to the points in Figure 1a. For example, in Figure 1b, the spline goes below the horizontal line, even though the self-serving points are above the control points in Figure 1a. This may confuse readers. A simple fix is to flip the y-axis to capture “Self-serving - Control” instead.

• Four decimal places are used throughout – this is distracting and probably not necessary. Two decimal places would suffice.

In summary, this was a clear and well-described paper that will contribute to the literature on default effects. Please let me know if you have any questions about this review.

Review signed: Scott Claessens (scott.claessens@gmail.com)

References

Berger, S., Kilchenmann, A., Lenz, O., Ockenfels, A., Schlöder, F., & Wyss, A. M. (2022). Large but diminishing effects of climate action nudges under rising costs. Nature Human Behaviour, 6(10), 1381-1385.

Chaudhuri, A., & Paichayontvijit, T. (2017). On the long-run efficacy of punishments and recommendations in a laboratory public goods game. Scientific Reports, 7(1), 12286.

Fehr, E., & Gächter, S. (2000). Cooperation and punishment in public goods experiments. American Economic Review, 90(4), 980-994.

Manganari, E., Mourelatos, E., Michos, N., & Dimara, E. (2022). Harnessing the power of defaults now and forever? The effects of mood and personality. International Journal of Electronic Commerce, 26(4), 472-496.

McCoy, C. E. (2017). Understanding the intention-to-treat principle in randomized controlled trials. Western Journal of Emergency Medicine, 18(6), 1075.

Ortmann, A., Ryvkin, D., Wilkening, T., & Zhang, J. (2023). Defaults and cognitive effort. Journal of Economic Behavior & Organization, 212, 1-19.

Zucchelli, M. M., Gambetti, E., Giusberti, F., & Nori, R. (2024). Use of default option nudge and individual differences in everyday life decisions. Cognitive Processing, 25(1), 75-88.

Reviewer #2: Summary:

The paper reports the results of an experiment conducted on Prolific testing the effects of a pro-social and a self-serving default value for extraction decisions in a Common Pool Resource Dilemma (CPRD) game compared to a control condition without any default. The default was implemented simply as a value pre-selected in a drop-down menu of possible decisions, which participants could change easily if desired. The results indicate significant default effects compared to the control condition, with an interesting asymmetry: the self-serving default led to significantly higher extraction decisions in all five rounds of its implementation, whereas the pro-social default only led to lower extraction decisions in the first three rounds. Both types of defaults did not lead to any significant spillover effects after their removal (i.e., in rounds 6-10). Another interesting finding is that the pro-social default leads participants with individualistic or competitive social preferences (as measured by the SVO measure) to extract less (whereas it has little effect on participants with cooperative and altruistic social preferences). Similarly, the self-serving default leads participants with cooperative or altruistic social preferences to extract more (with smaller effects on participants of individualistic or competitive type). Finally, the paper also finds that risk-averse participants are more strongly influenced by a default.

General Comments:

This is a well-written paper reporting the results of a well-designed and conducted experiment. The results are interesting, especially the analysis of heterogeneity in participants’ reactions to defaults (as a function of social preferences and risk aversion).

I am therefore in general quite positive about the paper; however, I do have some comments and questions that the authors should address in a revision.

Major Comments:

My major comments mainly refer to the reporting of the results. There is not always all relevant information provided to be able to understand and judge your analyses. Here are the specific points that I think should be addressed:

1. Did you use group averages in your analyses, or did you adjust for the non-independence of observations within the groups of four via clustering of standard errors? As participants interacted in groups of four and learned about each other’s extraction decisions, the independent level of observation is the group of four (unless maybe in the very first round of the CPRD game). Please explain how you handled this, especially in your mixed effects model where you analyze the interactions between individual difference variables (SVO and risk preferences) and the experimental treatments.

2. What do the p-values in square brackets ([…]) refer to? Is this the cluster-robust p-value? Or is this the p-value adjusted for multiple hypotheses testing? Please explain clearly how these p-values were derived.

3. In your pre-registration, you said that you would “account for multiple hypothesis testing by doing a correction on the tests such as Bonferroni correction”. You do not seem to have done that in your analyses (at least it is not mentioned in the paper). Please explain whether (and how exactly) you adjusted for multiple hypotheses testing, or, if not, why you chose not to.

Minor Comments:

4. P. 2 you write: “For instance, an electricity provider can offer renewable energy for consumers by default, promoting a course of action supported by policy or a prevailing social norm. Yet, some consumers may prefer other sources of energy and adapt their consumption habits to accommodate their individual preferences [16, 17]. People who are particularly insensitive to the collective issue of energy over-consumption may be more likely to override a default…”. This makes it sound like the possibility to override defaults that do not correspond with personal preferences is a negative outcome. From the standpoint of libertarian paternalism (e.g., Thaler & Sunstein, 2003), however, this is one of the great things about defaults: people with preferences different from the choice architect’s objective are not forced to just succumb to the default, but can opt out according to their own preferences. Indeed, one of the undesirable side-effects of green defaults may be that not enough people opt out when their personal preferences differ (see Ghesla et al., 2020; I believe this is also the paper that should be cited as current reference 38).

5. While I am myself positive about the potential of nudges or choice architecture interventions for behavioral change, when citing the paper by Mertens et al. (2022) on the effectiveness of nudges, I think it’s important to also cite the paper by Maier et al. (2022, PNAS) who, in a direct response to Mertens et al., argue that there is no meta-analytical evidence for the effectiveness of nudging after adjusting for publication bias.

6. You provide a screenshot of the CPRD decision screen in Figure 2, but despite mentioning it in line 210, I couldn’t find further instructions or screenshots for the CPRD in the supplementary material. Please provide these and ensure their location is clearly referenced.

7. I don’t understand the relevance of the following sentence in the context of why the authors elicited participants’ SVO and risk aversion: Line 131: “Moreover, certain decisions can pose greater challenges for specific individuals, such as the relatively slower evidence collection among equality seekers compared to categorical decision-makers [50].” Please elaborate or adapt this sentence for clarity.

8. Why did you measure SVO and risk attitude before the CPRD experiment? In principle, these measures could serve as primes and lead to undesirable spillovers into the CPRD game. You can argue that this is no major problem, because Tasks 1 and 2 were constant across conditions; however, one might also argue that such priming could interact with the treatments (e.g., if measuring SVO primes people to be more cooperative, this could lead them to react more positively to the pro-social default). Personally, I do not think it’s a major issue in the current experiment. However, in future work you might consider measuring such presumably stable individual preferences after the main experiment.

9. What is meant by “in the first rounds” in line 371? I assume you mean the first five rounds? If yes, please make this explicit. It is not completely clear, as you report an analysis for only rounds 3 to 5 just below on line 377.

10. Some typos or suggested language edits:

a. Please proof-read the paragraph on study design (starting line 148) as there are several minor mistakes or language issues.

b. Line 205: I would suggest to formulate this sentence less strongly: “mitigating selective attrition concerns” (instead of “discarding”).

c. Line 489: “a strategic incentive” (not “an strategic incentive”)

d. Line 530: “cost-efffective tool” (not “cost-effect tool”)

e. Line 530: “to ease” seems to be a strange choice of words. Please consider rephrasing this.

Congratulations to the authors on a nice paper and all the best for their future work!

References:

Ghesla, C., Grieder, M., & Schubert, R. (2020). Nudging the poor and the rich – A field study on the distributional effects of green electricity defaults. Energy Economics, 86, 104616.

Maier, M., Bartoš, F., Stanley, T. D., Shanks, D. R., Harris, A. J., & Wagenmakers, E. J. (2022). No evidence for nudging after adjusting for publication bias. Proceedings of the National Academy of Sciences, 119(31), e2200300119.

Thaler, R. H. and Sunstein, C. R. (2003). Libertarian paternalism. American Economic Review, 93(2):175-179.

Reviewer #3: Review of “From self-interest to collective action: The role of defaults in governing common resources"

This paper studies the effect of positive and negative defaults on behavior in a common pool resource lab experiment. It further correlates these behavioral effects with social preferences measured in terms of social value orientation (SVO) and risk attitudes. The study finds that these defaults have corresponding effects on initial behavior, which then converge to the Nash equilibrium over time, and faster for the positive default than the negative default. No spillover effects were detected after defaults were lifted in the second half of the experiment. The study is interesting and can potentially add to the literature on default effects. My comments are as follows.

1) Introduction/motivation: It is unclear how the renewable energy example on p.2 directly matched to the way defaults were experimentally implemented in terms of extraction values. There needs to be a better alignment of what the experiment does with what the paper claims it can tell us about which real-world applications. Clarifying the study’s domain of defaults that it concerns is important, considering that the framing and potentially behavior vary also with the specific defaults studied. Also, how does this study contribute to resolving the mixed results mentioned in p.3? I was puzzled by how this “work expands on this current understanding by testing the presence and nature of spillover effects when default values are lifted.” It was only later that I learned that the defaults were “lifted” in round 6 of 10 in all treatments, and this being a central feature of the experiment requires more emphasis and motivation in the introduction. Generally, I think the first section needs a careful rewrite to elucidate the contribution of the paper.

2) Experiment: While I understand the practicalities of excluding data from dropouts in the sample, and selecting those with high approval ratings, I wonder if this leads to a selection bias whereby certain types are self-excluded from the tests. It would be good to explain if this imposes any limitations on the findings and to explicitly defend the integrity of the experiment in this respect, if possible.

3) Analysis: I was unable to tell if the non-independence from repeated interaction was considered in the reported tests. For example, how was an independent observation defined in the non-parametric tests, and were clusters used in the multivariate tests? Also, the tests for behavior in the first and second halves of the experiment should be analyzed separately and the same applies for the presentation of their results and implications. On the matter of SVO types, are the distributions comparable with those of previous studies (related to my previous point 2) and across treatments? There were only three altruistic subjects so I would be extremely cautious about interpreting their “results”, while the present analysis (text and figures) treats them as prominently as dominant groups of individualists and cooperators. Section 4.3 mentions a hypothesis that was not previously developed.

4) The self-serving default has a noticeable negative effect on cooperation and convergence to the Nash equilibrium is slow, thus reflecting the dangers of such a default. This is also the case in the control, albeit to a lesser extent. In contrast, the pro-social default has the least negative effect on cooperation. This practical implication deserves more emphasis in the discussion.

Minor comments:

In the composite figures 1 and 3, the inconsistency of labels for A-D in caption and a-d in figure is a little confusing.

6. PLOS authors have the option to publish the peer review history of their article (what does this mean?). If published, this will include your full peer review and any attached files.

Reviewer #1: **Yes: **Scott Claessens

Reviewer #2: No

Reviewer #3: No

---

## [Author Response · Author response to Decision Letter 1]

21 May 2025

Response from the authors

We thank the reviewers and the editor for the comments and concerns raised, this will make the results more robust and the paper more clear, we appreciate the detailed analysis and interest in our work. We responded to every comment made by the reviewers in this document in blue letters and the changes reflected in red in the manuscript.

Reviewer #1:

In this paper, the authors tested the effects of prosocial and self-serving defaults on extraction levels in a common-pool resource dilemma. The results of an online economic game showed that prosocial defaults caused people to extract less from the common pool and self-serving defaults caused people to extract more. However, these effects were short-lived, disappearing entirely when defaults were removed halfway through the game. The effects were also moderated by Social Value Orientation and risk preferences: for example, the authors found that prosocial defaults were more effective for individualists and risk-averse participants.

The study was described very clearly and the results were easy to follow. I really liked the statistical approach taken in the paper – more studies with repeated games should use GAMs to track changes across rounds! I also assessed the pre-registration, data, and analysis code as part of my review. The pre-registration was comprehensive and clearly laid out the key hypotheses. I was able to reproduce the statistical results of the paper with the data and analysis code (after some changes to file paths in the R file).

We appreciate the thorough analysis and interest that the reviewer has in our work. Below, we respond to each comment in blue and provide the location in the manuscript where the changes can be found and highlighted in red letters.

That said, I had a few concerns about the paper, mainly related to the discussion of prior literature, the statistical analyses, and the treatment of dropouts.

First, I felt that the discussion of prior literature was scant in places, making it difficult to assess how much of a contribution this paper is making. For example, in the Introduction section, the authors could have discussed how previous psychological work has already studied how the effects of defaults vary depending on individual differences in anxiety and avoidance (Zucchelli et al. 2024), cognitive effort (Ortmann et al. 2023), and mood and personality (Manganari et al. 2022). In the Discussion section, it might also be worth discussing how the results are in line with other studies in behavioural economics, particularly studies showing that the effects of other interventions in repeated games often decay over time (e.g., Chaudhuri & Paichayontvijit, 2017) and go away entirely when removed (e.g., Fehr & Gächter, 2000). Regarding generalising the results to real-world settings, the authors could discuss the use of green defaults in airline carbon offsetting (Berger et al. 2022) and speculate as to whether such interventions might be more effective for more individualist and risk-averse customers. I am aware that the literature on defaults is very large, and this is not a review paper! But I think some extra discussion in places could better couch the current paper and its findings in the existing literature.

We thank the reviewer for taking the time and finding relevant literature for our paper. Indeed, this is not a review paper, but the sources provided can enrich our understanding of how defaults work and also give a solid background before presenting the results. We included the suggested references in the introduction and discussion sections of the paper. Find the changes in lines 121-127, 143-145 of the introduction and 550-552 and 559-560 in the discussion section.

Second, I had a few concerns with the analysis approach. While I liked the use of GAMs, the authors should consider including random intercepts for groups in the models in addition to the random intercepts for participants. The nested nature of the experiment means that there will be dependencies within groups. This group-level variation may not be fully captured by the “previous group extraction” fixed effect in the model. Do the results hold when this group-level variation is accounted for with additional random intercepts?

We thank the reviewer for this relevant observation. We agree that the random intercepts for our regression could capture the group-level variation. For this, we ran the GAM again, this time with a random effect for each group. You can find the results of the GAM in Table S8. We found that the results still hold, as the new random effect for the group-level is not significant. We added the new results, as well, we updated the formula in the Methods to include the new random effect variable.

Third, there is the perennial issue of dropouts in online experiments. If I’m understanding correctly, the authors’ pre-registered approach was to exclude any participants who dropped out and any groups containing dropouts. But this is problematic from a causal inference perspective, especially if dropouts are more common in one treatment group compared to others – if this is the case, it becomes difficult to know whether differences between experimental conditions are due to the treatment itself or due to differences in attentiveness and characteristics of participants who drop out. The authors could remedy this in two ways: (1) by analysing whether dropout rates vary systematically across treatment groups, and (2) by running an “intention-to-treat” analysis (McCoy, 2017) that fits the same models but retains observations from groups that contain dropouts. Since there are only a few dropouts, the results will likely not differ all that much, but this analysis would show that the main causal conclusions are unbiased.

We thank the reviewer for the remark. Dropouts in behavioural economics can be divisive. While we initially assessed the impact of dropouts in the first two tasks, we did not consider their effect in the third task. To test this, we performed three different tests:

We fitted a logistic regression to make sure that no individual factor (such as demographics and results of the previous two tasks) was predicting individual drop out. None of the variables we considered was significant, which confirms that drop out was not linked with individual characteristics. Find the results in Table S2 in the Supporting Information.

Addressing your suggestions, we conducted a chi-square test to assess whether dropout rates differ significantly across treatments, and found no significant differences. Find this also in lines 428-432.

Finally, we re-fitted the GAM under an intention-to-treat framework by including dropouts and the remaining members of their groups. The main findings remain consistent, although some coefficients differ between the full-sample model and the one restricted to completed participants. The results are reported in Table S9.

Finally, here are some minor additional points to consider:

• Some visual examples of “free-riding” and “cooperation” in the economic game might be helpful for readers, especially those outside of behavioural economics.

Thank you for this comment, we included some examples of water use and grazing in lines 32-36

• I am assuming that the treatment is at the group-level, such that all participants within a group experience the same treatment. If this is the case, it should be clearly stated in the Methods section.

You are right, thank you for pointing this out. We added a clarification in the Methods section, in lines 263-264.

• Figures 1b and 1c are slightly confusing because the splines go in the opposite direction to the points in Figure 1a. For example, in Figure 1b, the spline goes below the horizontal line, even though the self-serving points are above the control points in Figure 1a. This may confuse readers. A simple fix is to flip the y-axis to capture “Self-serving - Control” instead.

Thank you for this comment, we are assuming you mean Figures 3b and 3c. Indeed, we were torn between showing “Control-Self-serving”, or to show “Self-serving-Control”. We decided to change both Figures 3b and 3c to keep consistency of “Treatment vs Control”.

• Four decimal places are used throughout – this is distracting and probably not necessary. Two decimal places would suffice.

Thanks for this observation, we removed the extra decimal places throughout the document, with the exception of the p-values, that the minimum is set to be at p = 0.001.

In summary, this was a clear and well-described paper that will contribute to the literature on default effects. Please let me know if you have any questions about this review.

Review signed: Scott Claessens (scott.claessens@gmail.com)

References

Berger, S., Kilchenmann, A., Lenz, O., Ockenfels, A., Schlöder, F., & Wyss, A. M. (2022). Large but diminishing effects of climate action nudges under rising costs. Nature Human Behaviour, 6(10), 1381-1385.

Chaudhuri, A., & Paichayontvijit, T. (2017). On the long-run efficacy of punishments and recommendations in a laboratory public goods game. Scientific Reports, 7(1), 12286.

Fehr, E., & Gächter, S. (2000). Cooperation and punishment in public goods experiments. American Economic Review, 90(4), 980-994.

Manganari, E., Mourelatos, E., Michos, N., & Dimara, E. (2022). Harnessing the power of defaults now and forever? The effects of mood and personality. International Journal of Electronic Commerce, 26(4), 472-496.

McCoy, C. E. (2017). Understanding the intention-to-treat principle in randomized controlled trials. Western Journal of Emergency Medicine, 18(6), 1075.

Ortmann, A., Ryvkin, D., Wilkening, T., & Zhang, J. (2023). Defaults and cognitive effort. Journal of Economic Behavior & Organization, 212, 1-19.

Zucchelli, M. M., Gambetti, E., Giusberti, F., & Nori, R. (2024). Use of default option nudge and individual differences in everyday life decisions. Cognitive Processing, 25(1), 75-88.

Reviewer #2:

Summary:

The paper reports the results of an experiment conducted on Prolific testing the effects of a pro-social and a self-serving default value for extraction decisions in a Common Pool Resource Dilemma (CPRD) game compared to a control condition without any default. The default was implemented simply as a value pre-selected in a drop-down menu of possible decisions, which participants could change easily if desired. The results indicate significant default effects compared to the control condition, with an interesting asymmetry: the self-serving default led to significantly higher extraction decisions in all five rounds of its implementation, whereas the pro-social default only led to lower extraction decisions in the first three rounds. Both types of defaults did not lead to any significant spillover effects after their removal (i.e., in rounds 6-10). Another interesting finding is that the pro-social default leads participants with individualistic or competitive social preferences (as measured by the SVO measure) to extract less (whereas it has little effect on participants with cooperative and altruistic social preferences). Similarly, the self-serving default leads participants with cooperative or altruistic social preferences to extract more (with smaller effects on participants of individualistic or competitive type). Finally, the paper also finds that risk-averse participants are more strongly influenced by a default.

We thank the reviewer for the thorough feedback of our manuscript. Find the comments in blue after each point and the corrections in the manuscript can be found in red letters.

General Comments:

This is a well-written paper reporting the results of a well-designed and conducted experiment. The results are interesting, especially the analysis of heterogeneity in participants’ reactions to defaults (as a function of social preferences and risk aversion).

I am therefore in general quite positive about the paper; however, I do have some comments and questions that the authors should address in a revision.

Major Comments:

My major comments mainly refer to the reporting of the results. There is not always all relevant information provided to be able to understand and judge your analyses. Here are the specific points that I think should be addressed:

1. Did you use group averages in your analyses, or did you adjust for the non-independence of observations within the groups of four via clustering of standard errors? As participants interacted in groups of four and learned about each other’s extraction decisions, the independent level of observation is the group of four (unless maybe in the very first round of the CPRD game). Please explain how you handled this, especially in your mixed effects model where you analyze the interactions between individual difference variables (SVO and risk preferences) and the experimental treatments.

We thank the reviewer for bringing this important point to our attention, which was also raised by the other reviewers. Indeed, we accounted for group interaction in our mixed effects model, but we included it in the Supporting Information, not in the main text nor in the mixed effects model formula. We added random intercepts for groups in addition to participants in the GAM. Given the nested design, we tested this adjustment and found the results unchanged, with the group-level random effect being non-significant. We changed this in the Methods section in lines 396-403 and added the table below (see Table S8 in the Supporting Information).

2. What do the p-values in square brackets ([…]) refer to? Is this the cluster-robust p-value? Or is this the p-value adjusted for multiple hypotheses testing? Please explain clearly how these p-values were derived.

We thank the reviewer for pointing this out. Indeed, initially, we included the p-values in brackets to account for the multiple hypotheses testing. We stated five different hypotheses in our pre-registration, and the p-values in brackets represented this adjustment. However, as we stated in the next question, we decided to deviate from this correction for multiple hypotheses (see the answer of question 3). We stated this more clearly in the Supporting Information section, lines 935-942 for clarification, and we removed the squared brackets all over the document.

3. In your pre-registration, you said that you would “account for multiple hypothesis testing by doing a correction on the tests such as Bonferroni correction”. You do not seem to have done that in your analyses (at least it is not mentioned in the paper). Please explain whether (and how exactly) you adjusted for multiple hypotheses testing, or, if not, why you chose not to.

Thanks again for bringing up this important point. While our pre-registration mentioned a multiple testing correction, we did not apply one, as our hypotheses were pre-specified and limited in number. Following (Armstrong, 2014), such corrections are primarily warranted in exploratory analyses with many unplanned tests; in confirmatory settings, they risk inflating Type II errors. Since we defined the hypotheses and corresponding tests in advance, rather than exploring the data post hoc or conducting a large number of unplanned comparisons, we considered the risk of inflated Type I error to be limited. Nevertheless, we acknowledge the deviation from the pre-registration and have now noted this in the manuscript in lines 935-942.

Minor Comments:

4. P. 2 you write: “For instance, an electricity provider can offer renewable energy for consumers by default, promoting a course of action supported by policy or a prevailing social norm. Yet, some consumers may prefer other sources of energy and adapt their consumption habits to accommodate their individual preferences [16, 17]. People who are particularly insensitive to the collective issue of energy over-consumption may be more likely to override a default…”. This makes it sound like the possibility to override defaults that do not correspond with personal preferences is a negative outcome. From the standpoint of libertarian paternalism (e.g., Thaler & Sunstein, 2003), however, this is one of the great things about defaults: people with preferences different fr

---

## [Decision Letter · Decision Letter 1]

4 Jul 2025

PONE-D-25-04274R1From self-interest to collective action: The role of defaults in governing common resourcesPLOS ONE

Dear Dr. Montero-Porras,

Thank you for submitting your manuscript to PLOS ONE. After careful consideration, we feel that it has merit but does not fully meet PLOS ONE’s publication criteria as it currently stands. Therefore, we invite you to submit a revised version of the manuscript that addresses the points raised during the review process.

Please see the comments from the previous reviewers. I also read your revision and thought that major problems were solved. The reviewers pointed out several minor problems.Please address them and provide point-by-point responses.

We look forward to receiving your revised manuscript.

Kind regards,

Yutaka Horita

Academic Editor

PLOS ONE

Journal Requirements:

Reviewers' comments:

Reviewer's Responses to Questions

**Comments to the Author**

1. If the authors have adequately addressed your comments raised in a previous round of review and you feel that this manuscript is now acceptable for publication, you may indicate that here to bypass the “Comments to the Author” section, enter your conflict of interest statement in the “Confidential to Editor” section, and submit your "Accept" recommendation.

Reviewer #1: (No Response)

Reviewer #2: (No Response)

2. Is the manuscript technically sound, and do the data support the conclusions?

Reviewer #1: Yes

Reviewer #2: Yes

3. Has the statistical analysis been performed appropriately and rigorously? 

Reviewer #1: Yes

Reviewer #2: Yes

4. Have the authors made all data underlying the findings in their manuscript fully available?

Reviewer #1: Yes

Reviewer #2: Yes

5. Is the manuscript presented in an intelligible fashion and written in standard English?

Reviewer #1: Yes

Reviewer #2: Yes

6. Review Comments to the Author

Reviewer #1: Thank you to the authors for dealing with all of the points that I raised in my previous review. I am happy to recommend this paper for publication at PLOS One. I hope that it will make a positive contribution to the literature on default effects.

There is one remaining minor issue that should be addressed before publication. It looks like the logistic regression testing the associations between demographics and drop out rates (reported in Table S2) may not have converged properly, since the standard errors for many of the parameters are huge. This may be because there are only a small number of dropouts in the dataset. The authors should check whether the results of this model are valid.

Reviewer #2: I would like to thank the authors for the revision, the changes they made to the paper, and their responses to my first-round comments. Most of my comments have been addressed. I have the following remaining comments relating to points already raised in the first review.

1. In your response letter, you explain that you’ve accounted for non-independence in your mixed-effects model by including random intercepts for groups, which is appropriate. As a robustness check, however, I encourage you to report cluster-robust standard errors at the group level to show that your key ppp-values remain unchanged even if the random effect is negligible. Additionally, your reply does not address how you handled non-independence in the non-parametric tests and ANOVAs reported in the Results section. Please specify on how many observations each of these tests is based, and describe what adjustments (e.g., aggregation to group means or clustering corrections) you applied to account for within-group dependence.

2. I see no major problems with your deviations from the pre-registration as outlined in S0.1.1 in the Supporting Information. However, I think you should at least put a reference to this section in the main body of the paper (e.g., on line 213, when mentioning the pre-registration).

3. Reference #47 should be changed to the following: Ghesla, C., Grieder, M., & Schubert, R. (2020). Nudging the poor and the rich – A field study on the distributional effects of green electricity defaults. Energy Economics, 86, 104616.

7. PLOS authors have the option to publish the peer review history of their article (what does this mean?). If published, this will include your full peer review and any attached files.

Reviewer #1: **Yes: **Scott Claessens

Reviewer #2: No

---

## [Author Response · Author response to Decision Letter 2]

5 Aug 2025

Response to reviewers

We thank the reviewers and the editor for the comments and concerns raised, this will make the results more robust and the paper more clear, we appreciate the detailed analysis and interest in our work. We responded to every comment made by the reviewers in this document in blue letters and the changes reflected in red in the manuscript.

Reviewer #1:

Thank you to the authors for dealing with all of the points that I raised in my previous review. I am happy to recommend this paper for publication at PLOS One. I hope that it will make a positive contribution to the literature on default effects.

There is one remaining minor issue that should be addressed before publication. It looks like the logistic regression testing the associations between demographics and drop out rates (reported in Table S2) may not have converged properly, since the standard errors for many of the parameters are huge. This may be because there are only a small number of dropouts in the dataset. The authors should check whether the results of this model are valid.

We thank the reviewer for carefully checking our previous edits and responses. Regarding Table S2, we agree that several variables (such as nationality and age) exhibit large standard errors. This is primarily due to (1) the small number of dropouts (21 out of 709 participants) and (2) the limited contribution of these variables to the binomial regression. This supports our argument that neither demographic factors nor prior task performance significantly influenced participant dropout.

To further examine this, we fitted a reduced logit regression including only task-related variables (SVO score and gamble choice). We found no significant association between SVO score or gamble choice and dropout (see Table S3). We thank the reviewer for prompting this analysis, which led to an additional insight into participant behaviour on Prolific. Find the changes in lines 258-261.

Reviewer #2:

I would like to thank the authors for the revision, the changes they made to the paper, and their responses to my first-round comments. Most of my comments have been addressed. I have the following remaining comments relating to points already raised in the first review.

1. In your response letter, you explain that you’ve accounted for non-independence in your mixed-effects model by including random intercepts for groups, which is appropriate. As a robustness check, however, I encourage you to report cluster-robust standard errors at the group level to show that your key ppp-values remain unchanged even if the random effect is negligible.

We thank the reviewer for pointing this out, We used robust standard errors clustered at the group level in our GAM. When accounting for potential intra-group correlation, the initial parameters (e.g. Treatment, SVO score and gamble choice smoothed by round) remained significant (see Supplementary Table S11), confirming the robustness to non-independence of observations within a group. We mentioned this in the text in lines 521-526.

Additionally, your reply does not address how you handled non-independence in the non-parametric tests and ANOVAs reported in the Results section. Please specify on how many observations each of these tests is based, and describe what adjustments (e.g., aggregation to group means or clustering corrections) you applied to account for within-group dependence.

We thank the reviewer for this comment. We changed the statistical tests used (ANOVA’s and KS tests) to account for the non-independence of the observations by computing group averages across rounds, and made our tests on these averages. We find that all statistical tests hold but one, where we state that extractions in the Self-serving treatment by Risk-seekers participants were higher than Risk-averse ones. Despite this, the main results of our paper hold, as shown with the GAM regressions.

Moreover, to make sure that those statistical tests are still robust, we fitted the relevant ANOVA’s taking only the first round, where behaviour cannot yet be correlated within a group. We also find that the results still hold, but one, where we state that Risk-averse participants extracted less in the Pro-social treatment than Risk-seeking ones. Find this analysis in Section S0.8.

We added this new information about the analysis in Section S0.1.1 (Deviation from the pre-registration).

2. I see no major problems with your deviations from the pre-registration as outlined in S0.1.1 in the Supporting Information. However, I think you should at least put a reference to this section in the main body of the paper (e.g., on line 213, when mentioning the pre-registration).

Thanks for pointing this out. In our text, when we mention the OSF Preregistration we also point to the deviations section in our Supplementary Information. Find the changes in lines 213-215.

3. Reference #47 should be changed to the following: Ghesla, C., Grieder, M., & Schubert, R. (2020). Nudging the poor and the rich – A field study on the distributional effects of green electricity defaults. Energy Economics, 86, 104616.

Thank you for this, perhaps we pointed out to the wrong reference of Ghesla, the one in line 74 should be updated and it’s now the reference number 22.

---

## [Decision Letter · Decision Letter 2]

15 Aug 2025

From self-interest to collective action: The role of defaults in governing common resources

PONE-D-25-04274R2

Dear Dr. Montero-Porras,

We’re pleased to inform you that your manuscript has been judged scientifically suitable for publication and will be formally accepted for publication once it meets all outstanding technical requirements.

Kind regards,

Yutaka Horita

Academic Editor

PLOS ONE

Additional Editor Comments (optional):

Reviewers' comments:

Reviewer's Responses to Questions

**Comments to the Author**

1. If the authors have adequately addressed your comments raised in a previous round of review and you feel that this manuscript is now acceptable for publication, you may indicate that here to bypass the “Comments to the Author” section, enter your conflict of interest statement in the “Confidential to Editor” section, and submit your "Accept" recommendation.

Reviewer #1: All comments have been addressed

Reviewer #2: All comments have been addressed

2. Is the manuscript technically sound, and do the data support the conclusions?

Reviewer #1: Yes

Reviewer #2: Yes

3. Has the statistical analysis been performed appropriately and rigorously? 

Reviewer #1: Yes

Reviewer #2: Yes

4. Have the authors made all data underlying the findings in their manuscript fully available?

Reviewer #1: Yes

Reviewer #2: Yes

5. Is the manuscript presented in an intelligible fashion and written in standard English?

Reviewer #1: Yes

Reviewer #2: Yes

6. Review Comments to the Author

Reviewer #1: (No Response)

Reviewer #2: (No Response)

7. PLOS authors have the option to publish the peer review history of their article (what does this mean?). If published, this will include your full peer review and any attached files.

Reviewer #1: **Yes: **Scott Claessens

Reviewer #2: No

---

## [Editor Report · Acceptance letter]

PONE-D-25-04274R2

PLOS ONE

Dear Dr. Montero-Porras,

I'm pleased to inform you that your manuscript has been deemed suitable for publication in PLOS ONE. Congratulations! Your manuscript is now being handed over to our production team.

Kind regards,

on behalf of

Dr. Yutaka Horita

Academic Editor

PLOS ONE